# The Effects of Frequent Coffee Drinking on Female-Dominated Healthcare Workers Experiencing Musculoskeletal Pain and a Lack of Sleep

**DOI:** 10.3390/jpm13010025

**Published:** 2022-12-22

**Authors:** Yong-Hsin Chen, Ying-Hsiang Chou, Tsung-Yuan Yang, Gwo-Ping Jong

**Affiliations:** 1Department of Public Health, Chung Shan Medical University, Taichung 402, Taiwan; 2Department of Occupational Safety and Health, Chung Shan Medical University Hospital, Taichung 402, Taiwan; 3Department of Medical Imaging and Radiological Sciences, Chung Shan Medical University, Taichung 402, Taiwan; 4Department of Radiation Oncology, Chung Shan Medical University Hospital, Taichung 402, Taiwan; 5Department of Internal Medicine, Chung Shan Medical University Hospital, Taichung 402, Taiwan; 6Institute of Medicine, College of Medicine, Chung Shan Medical University, Taichung 402, Taiwan

**Keywords:** coffee, musculoskeletal pain, sleep, Nordic musculoskeletal questionnaire, neck and shoulder pain

## Abstract

Previous research has demonstrated that chronic diseases can occur due to musculoskeletal (MS) pain and poor sleep. It is also worth noting that the caffeine in coffee can reduce overall sleep duration, efficiency, and quality. Thus, the present study examines the effects of frequent coffee drinking (two cups per day) on individuals experiencing MS pain and a lack of sleep during the COVID-19 period. This observational and cross-sectional study recruited 1615 individuals who completed the self-reported (Nordic musculoskeletal) questionnaire. Long-term, frequent coffee drinking and a sleep duration of less than 6 h per day were significantly associated with neck and shoulder pain among healthy individuals. The mediation model demonstrated that the shorter sleep duration and drinking multiple cups of coffee per day had a two-way relationship that worsened such pain over the long term. Specifically, individuals who experienced such pain frequently drank multiple cups of coffee per day, which, in turn, shortened their sleep durations. In summary, long-term coffee drinking creates a vicious cycle between MS pain and sleep duration. Therefore, the amount of coffee should be fewer than two cups per day for individuals who sleep less than 6 h per day or suffer from MS pain, especially neck and shoulder pain.

## 1. Introduction

Coffee is one of the most popular beverages for people of many ages. It is a complex chemical mixture that contains caffeine, which is a purine alkaloid that is naturally found in coffee beans [1] and contributes to its bitterness [2]. Caffeine stimulates the central nervous system, which can increase alertness, blood circulation, and respiration [3]. However, caffeine has biphasic effects, i.e., lower doses can provide some behavioral stimulation, whereas higher doses can lead to anxiety, aversion, irritability, and discomfort [4]. Despite clinical studies demonstrating the adjuvant analgesic effects of caffeine [5], long-term coffee drinking can negatively affect health and musculoskeletal (MS) pain. In health, individuals drinking more than five cups of coffee per day can have an increased risk of myocardial infarction or unstable angina [6]. In MS pain, related research showed that drinking more than seven cups of coffee per day was associated with a higher risk of knee osteoarthritis among Korean men [7]. Interestingly, patients with chronic back pain tend to drink two times as much caffeine as those without such pain [8], whereas individuals with chronic daily headaches were generally high caffeine consumers before the onset of such headaches [9].

MS pain is common in many occupations, and it is one of the main reasons for long-term sick leave [10]. In the United States, 13% of the total workforce experienced a loss caused by body pains, with lost productive time costs estimated at USD 61.2 billion annually [11]. Although different occupations can affect MS pain at various anatomical sites and have diverse risk factors [12], a recent study in the Netherlands showed that the top three self-reported MS pains include lower-back pain, shoulder pain, and neck pain [13]. In addition, previous studies have demonstrated that work hours [14,15], occupational stress [16,17], alcohol consumption [18,19,20,21], sleep duration [22,23,24], exercise habits [25], and chronic diseases [26,27] contribute to MS pain.

Poor sleep quality is a common health problem among medical staff [28,29]. Reduced sleep duration and poor sleep quality have become more common during the past decades [30], leading to poor health outcomes [31] and even increased mortality [32]. Despite the recommended minimum sleep duration of 7 h per night for healthy adults, only 25% of adults achieve this amount [33]. Notably, lack of sleep can lead to impaired daytime function [34], increased occupational injury [35], and reduced productivity [36].

Overall, a close relationship was found between sleep and MS pain. For instance, because sleep problems can significantly reduce pain tolerance [37], individuals with chronic pain are more likely to experience insomnia [38]. Caffeine in coffee can also reduce total sleep duration, efficiency, and quality [39]. In addition, frequent consumption of caffeinated drinks can negatively affect habitual sleep duration [40].

From a micro and physiological perspective, adenosine is a purine nucleoside and a ubiquitous endogenous neurotransmitter that signals through four receptors (A1R, A2AR, A2BR, and A3R) in the brain to inhibit arousal and increase drowsiness [41]. Among these four receptors, A1R may be related to pain-sensing neurons [42]. Some evidence has demonstrated that A1R activation can produce antinociception of postoperative [43], neuropathic [44], and inflammatory [45] pain. In this regard, one study of mice found that acupuncture causes the release of nucleotides and adenosine to relieve pain [46]. However, these antinociceptive effects can be blocked by caffeine [47]. Notably, individuals with chronic insomnia were found to have reduced adenosine [48]. Moreover, impaired sleep significantly increases the risk of reduced pain tolerance [39]. These results suggest that the effects of caffeine on adenosine could play a pivotal role in pain development. Based on previous research, we propose the following hypotheses:

**Hypothesis** **1:***Coffee intake is significantly associated with increased risk of MS pain*.

**Hypothesis** **2:***Individuals with shorter sleep durations are more susceptible to MS pain*.

**Hypothesis** **3:***Coffee intake could lead to a vicious circle between lack of sleep and MS pain*.

## 2. Materials and Methods

This observational and cross-sectional study was initially conducted from a hospital affiliated with a medical university in Taichung, Taiwan, from March to April 2021. All 2531 healthcare workers who had served for one year in the hospital were distributed a QR code for a Google Forms-linked questionnaire by email. Among them, 1633 (64.52%) individuals completed the self-reported questionnaire, after which 1615 (63.81%) were deemed valid after those with missing data were excluded. Specifically, we used questionnaires, including the Nordic musculoskeletal questionnaire (NMQ), to obtain the participants’ basic demographic variables, family factors, living habits, work, physical health, and MS pain. The study protocol was approved by the Institutional Review Board of Chung Shan Medical University Hospital on 25 August 2021 (No: CS1-21108).

This study adopted the NMQ, modified and translated by the Taiwan Institute of Occupational Safety and Health [49], to survey the presence of pain attributable to work-related factors in the preceding year. The pain sites on the NMQ were classified as the neck, left or right shoulder, upper back, waist or lower back, left or right elbow, left or right wrist, left hip/thigh/buttock, right hip/thigh/buttock, left or right knee, and left or right ankle. The options for the frequency of each pain site were every day, once a week, once a month, once every half a year, and at least once every half a year, scored as 100, 80, 60, 40, and 20 points, respectively. Factor analysis was also adopted in the NMQ to determine the underlying variables that could effectively explain most of the questionnaire items. Through varimax rotation, the standardized scoring coefficients constituted new factor loadings and were defined according to their corresponding significance. The new factors that featured vector values exceeding 1 were retained according to the principle proposed by Hair et al. [50].

In the questionnaire, the basic response options included male or female for gender; age; “married” or “other” for marriage; and “without child,” “one child,” “two children,” “three children,” and “more than three children” for having children. The survey also asked if the participants engaged in leisure activities with family/friends during vacation time. The response options included “always,” “often,” “sometimes,” “seldom,” and “never.” Regarding their education, the response options were “master’s degree or above” and “university degree or below,” while the response options for self-reported sleep duration per day included “less than 5 h,” “between 5 and 6 h,” “between 6 and 7 h,” “between 7 and 8 h,” and “more than 8 h.” As for their coffee intake per day, the response options were “more than 2 cups per day,” “2 cups per day,” “1 cup per day,” “occasionally,” and “never.” Regarding their alcohol use, the response options included “alcohol use in a month” and “no alcohol use in a month,” while the response options were “yes” and “no” for exercising at least once a week. As for their overtime work, the response options were “seldom,” “fewer than 45 h per month,” “45 to 80 h per month,” and “more than 80 h per month,” while “irregular,” “regular,” “night,” and “day” were the response options for shift schedules. Finally, the participants were classified as physicians, nurses, professional and technical personnel, and administrative staff. They were also asked about the presence of chronic diseases. In this regard, the presence of one or more diseases was classified as a “yes” response.

Regarding the statistical methods, factor analysis [50] was adopted for the NMQ to determine new underlying variables, while a *t*-test or one-way ANOVA was adopted to examine the differences between the continuous variables. Additionally, a chi-square test or Fisher’s exact test was conducted to determine the significant differences in the categorical variables, while simple/multiple linear or logistic regression was used to examine the correlation between the dependent variable (DV) and the independent variable (IV), in the absence (or presence) of the controlled variables. The mediation effects among the IV, DV, and mediator were based on the following strategy proposed by Baron and Kenny [51]: 1) in the presence of the first-stage effect, the IV significantly affects the mediation factor; 2) in the absence of the mediation factor, the IV significantly affects the DV; 3) in the presence of the second-stage effect, the mediation factor has a significant effect on the DV; and 4) the effect of the IV on the DV weakens upon the addition of a mediation factor in the model.

A mediation model suitable for combining the categorical and continuous variables was developed by Iacobucci (2012) [52]. The formulas are as follows:

If the mediation factor and dependent variables are continuous variables, then the original formula of the Sobel test is applicable:$$Z=\frac{a\times b}{\sqrt{b^{2}s_{a}{}^{2}+a^{2}s_{b}{}^{2}}}$$

If the mediation factor or dependent variables are categorical variables, then the original formula of the Sobel test is rederived into a new formula:$$Z_{mediation\;}\left(Z_{m}\right)=\frac{\frac{a}{s_{a}}\times \frac{b}{s_{b}}}{\sqrt{{\left(\frac{a}{s_{a}}\right)}^{2}+{\left(\frac{b}{s_{b}}\right)}^{2}+1}}$$

Among them, a is the simple linear or logistic regression coefficient for the independent variable against the mediation factor, while b is the regression coefficient for the mediation factor against the dependent variable in the binary linear or logistic regression model. Additionally, $s_{a}$ and $s_{b}$ represent the standard deviations of a and b, respectively, while the results exceeding |1.96|, |2.57|, and |3.90| (for the two-tailed test) are significant at α = 0.05, 0.01, and 0.0001, respectively. In this study, the analyses were performed using SAS Enterprise Guide 7.1 software (SAS Institute Inc., Cary, NC, USA), and the significance was set at *p* < 0.05.

## 3. Results

Regarding the detailed description, the description of the basic demographics, sleep duration per day, and coffee intake of 1615 participants are shown in the Appendix A. The results demonstrated that marriage (*p* = 0.016), engaging in leisure activities with family/friends (*p* < 0.0001), coffee intake per day (*p* < 0.0001), exercise at least once a week (*p* = 0.008), overtime work in a month (*p* < 0.0001), shift schedules (*p* < 0.0001), and profession (*p* = 0.005) were associated with sleep duration per day. In addition, gender (*p* = 0.024), age (*p* < 0.0001), marriage (*p* < 0.0001), having children (*p* < 0.0001), education (*p* < 0.0001), alcohol use (*p* < 0.0001), exercise at least once a week (*p* = 0.002), and profession (*p* = 0.001) were related to coffee intake.

Table 1 illustrates that the common pain sites included both shoulders (43.09%), neck (36.22%), waist or lower back (27.93%), and upper back (16.90%). According to the principle proposed by Hair and Anderson (1995) [50], Factors 1 and 2 were retained because their vector values exceeded 1. In addition, the factor loadings were converted into standardized scoring coefficients through varimax rotation. The relatively large factor loading values for Factors 1 and 2 corresponded to the neck and both shoulder pain and both ankle pain sites, respectively. Thus, Factors 1 and 2 were redefined into two new variables: the neck and both shoulder pain (NBSP) score and the both ankle pain (BAP) score.

According to Table 2, there were significant differences in the NBSP scores for gender (*p* < 0.001), age (*p* = 0.003), marriage (*p* = 0.003), having children (*p* = 0.006), education (*p* = 0.034), sleep duration per day (*p* < 0.001), coffee intake per day (*p* = < 0.001), alcohol use (*p* = 0.001), exercise at least once a week (*p* = 0.001), overtime work per month (*p* < 0.0001), profession (*p* = 0.036), and suffering from chronic diseases (*p* < 0.0001). There were no significant differences in the BAP scores among the survey variables, except for education (*p* < 0.0001). Regarding the other survey variables, the females obtained higher NBSP scores than the males (0.04 ± 0.93 vs. −0.17 ± 0.84). Moreover, individuals who were 38–45 years of age (0.15 ± 0.96), were married (0.07 ± 0.96), were parents (0.07 ± 0.96), had a master’s degree or above (0.11 ± 0.99), had a sleep duration of less than 5 h (0.26 ± 1.04), drank more than two cups of coffee per day (0.61 ± 1.25), used alcohol in a month (0.10 ± 0.97), had no weekly exercise (0.09 ± 0.97), worked overtime more than 45 h per month (0.54 ± 1.35/0.44 ± 1.14), were nurses (0.08 ± 0.94), or suffered from chronic diseases (0.20 ± 1.03) achieved higher NBSP scores than the others. Simple multiple linear or logistic regression was also used to examine the correlation between the dependent and independent variables in the absence (or presence) of the controlled variables.

Since the number of individuals with a sleep duration of less than 5 h per day was only 63 (Table 2), the variables of sleep duration of less than 5 h per day and sleep duration between 5 and 6 h per day were combined into a new variable: sleep duration of less than 6 h per day (SLD < 6 h). Furthermore, since the number of individuals who drank more than two cups of coffee per day was only 26, the variables of drinking more than two cups of coffee per day or drinking just two cups of coffee per day were combined into a new variable: drinking multiple cups of coffee (MCC) per day.

As shown in Table 3, which presents the effects of sleep duration per day and drinking MCC per day on NBSP, drinking MCC per day was significantly associated with an increased level of NBSP in the simple or multiple linear regression models (B = 0.32, *p* = 0.001; 0.23, *p* = 0.016), while SLD < 6 h was significantly associated with an increased level of NBSP in the simple or multiple linear regression models (B = 0.21, *p* < 0.0001; 0.15, *p* = 0.001). The results in Table 3 confirm Hypotheses 1 and 2 (illustrated in the introduction), that coffee intake (more than two cups per day) and a shorter sleep duration (less than 6 h per day) are associated with MS pain (especially neck and both shoulders pain).

Finally, this study used mediation analysis to determine the existence of a mutual relationship between coffee, MS pain, and sleep. Figure 1.1 demonstrates that drinking MCC per day mediated the effect of SLD < 6 h on increased levels of NBSP (Z_m_ = 2.27, *p* < 0.05). Lack of sleep (<6 h) also caused individuals to drink more coffee per day, which led to more frequent neck and shoulder pain. Figure 1.2 illustrates that SLD < 6 h mediated the effect of drinking MCC per day on increased NBSP (Z_m_ = 2.95, *p* < 0.01). Overall, Figure 1 and Figure 2 demonstrate that long-term coffee drinking and lack of sleep can further increase the occurrence of neck and shoulder pain.

Figure 3 adopts SLD < 6 h and NBSP as dependent and independent variables for the mediation model, respectively. Based on the findings, drinking MCC per day mediated SLD < 6 h and increased NBSP (Z_m_ = 2.5, *p* < 0.05). Specifically, individuals who suffer from neck and should pain tend to drink coffee to cope with such pain. However, it eventually decreases their sleep duration per day. Figure 1, Figure 2 and Figure 3 confirmed Hypothesis 3 and determined that coffee intake (more than two cups per day) really opens the vicious circle between lack of sleep (less than 6 h per day) and MS pain (especially neck and shoulders pain).

## 4. Discussion

The present study confirms three hypotheses and determined that long-term heavy coffee intake (two cups per day) and a shorter sleep duration (<6 h per day) are associated with neck and shoulder pain. Notably, long-term heavy coffee intake plays a mediating factor in the vicious circle between shorter sleep duration and neck and shoulder pain. In addition, alcohol use, the lack of regular exercise at least once a week, overtime work in a month, and the presence of chronic diseases were significantly associated with pain in the neck/shoulders or ankles.

Related studies have illustrated that reduced alcohol use [18,19,20,21], physical activity [21], and fewer work hours [14] could reduce the risk of MS pain. In addition, individuals with chronic diseases [26,27] have a high risk for MS pain. These risk factors are consistent with our findings.

A literature review on healthcare workers demonstrated that MS pain occurred primarily in the lower and upper back, neck, and shoulders [53]. Our study found that the common pain sites were the shoulders (43.09%), neck (36.22%), waist or lower back (27.93%), and upper back (16.90%), which was consistent with the findings of a previous study.

Only 25% of adults achieve the recommended minimum sleep duration of 7 h per night for healthy adults [39]. However, only 16.72% of healthcare workers in the present study satisfy the 7 h sleep condition, as shown in Table 2. Therefore, the lack of sleep could be a common problem among healthcare workers in Taiwan, and this should be noted and further explored.

### 4.1. First Hypothesis: Coffee Intake Is Significantly Associated with an Increased Risk of MS Pain

Previous studies have found that individuals with a high caffeine intake (4–12 cups/day) had more severe pain than those with a low (0.25–1.5 cups/day) or moderate (2–3.5 cups/day) caffeine intake [54]. In addition, men who drink more than seven cups of coffee per day have an increased risk of knee osteoarthritis [7]. Our study determined that among healthy individuals, long-term drinking of more than two cups of coffee per day was associated with frequent neck and shoulder pain (Table 3, B = 0.23, *p* = 0.016). Since the half-life of caffeine is approximately 4 h [55], drinking two or more cups of coffee per day can reach the threshold of caffeine’s effect on MS pain, depending on one’s genetics [56]. Based on these results, we can confirm our first hypothesis.

### 4.2. Second Hypothesis: Individuals with Shorter Sleep Durations Are More Susceptible to MS Pain

Evidence suggests a close link between short sleep durations and impairments in several physiological responses, including pain [57]. A study on middle-aged adults in the U.S. demonstrated that a sleep duration of <6 h was associated with greater next-day pain [58]. In addition, individuals who reported >6 h of sleep were more likely to have improved pain conditions [59]. Our study found important evidence that sleep duration was associated with MS pain at specific sites. Individuals with sleep durations of <5 or 6 h tend to experience more neck and shoulder pain than others (Table 2; mean = 0.26 ± 1.04, 0.12 ± 1.00). Table 3 shows that a sleep duration of <6 h per day was significantly associated with increased neck and shoulder pain in the multiple regression model (B = 0.15, *p* = 0.001). These results confirm our second hypothesis.

### 4.3. Third Hypothesis: Coffee Intake Could Lead to a Vicious Circle between Lack of Sleep and MS Pain

A previous study showed that individuals who reported a sleep duration of <6 h consumed 3.6 times more caffeine per day than those who reported a sleep duration of >8 h [40]. Our study of healthcare workers found that individuals who reported a sleep duration of <6 h consumed 2.69 times (Figure 1.1, β = 0.99, odds ratio = e0.99 = 2.69, *p* < 0.0001) more caffeine per day than those who reported a sleep duration of >6 h. This close relationship between sleep duration and coffee intake indicates a causal relationship between sleep duration, NBSP, and coffee intake.

The mediation model in Figure 1.1 demonstrates that individuals who had shorter sleep durations tended to drink multiple cups of coffee, which can lead to increased MS pain (Zm = 2.27, *p* < 0.05). In addition, the mediation model in Figure 1.2 shows that individuals who chronically drink multiple cups of coffee generally experience shorter sleep durations and increased MS pain (Zm = 2.95, *p* < 0.01). These mediation models regarding coffee intake, MS pain, and sleep duration show that long-term heavy coffee intake (more than two cups per day) plays a mediating role in the two-way association of sleep duration <6 h and NBSP. Specifically, long-term heavy coffee intake will induce a vicious circle of sleep and neck and shoulder pain. These results are consistent with our third hypothesis.

This study has several limitations. First, we used the number of cups to measure the degree of caffeine intake per day. However, this is not an exact measurement method because cups have different volumes. Second, different coffee-brewing methods can lead to varied caffeine concentrations and errors in the dose–response of caffeine on MS pain. However, we believe that the differences in volume and caffeine concentration can be overcome. In addition, caffeine’s effect on MS pain reaches the threshold depending on one’s genetics [56]. Therefore, the threshold of more than two cups of coffee per day might not be suitable for other countries or races. Additionally, since sleep duration and sleep quality are subjective, future research should adopt other scales to measure sleep-related issues. Third, MS pain can be the result of workloads, work styles, or posture. Unfortunately, our study did not collect such data in the regression models. Fourth, the effects of caffeine on individuals can be associated with genetics [56] and nationality. For example, the effects of coffee intake on MS pain in Europeans or Americans may differ from our results because our participants were Taiwanese.

Despite the adjustment for sex in the multiple linear regression, the results of the present study could be better suited to women because female participants accounted for >80% of the study population. Regarding the sex difference in MS pain, it could be caused by estrogen and progesterone. For instance, testosterone, the major male sex hormone, protects men from chronic MS pain [60]. Because the study population only included physicians, nurses, professional and technical personnel, and administrative staff, we added “healthcare workers” in the title to limit the applicability to occupational groups.

Notably, we could determine whether high work stress or emotional exhaustion caused by the pandemic affected the findings; thus, a similar study during the nonpandemic period should be replicated, and its results compared with those from the pandemic period. Finally, the mediation models in our study could be biased [61] because the relationship was based on a higher risk of judgment. Therefore, we excluded the phrase “causal relationship” to avoid confusion.

## 5. Conclusions

The present study determines that keeping good living habits (such as decreased alcohol use, regular exercise a week, and sufficient sleep), maintaining physical health (such as staying away from chronic diseases), and avoiding overtime work are ways to lower the risk of MS pain. We further examined the effects of frequent coffee drinking on individuals experiencing MS pain and lack of sleep. Based on the results, neck and shoulder pain was the most common among the healthcare workers. In addition, a sleep duration of less than 6 h and drinking more than two cups of coffee per day increased the occurrence of such pain, while controlling for other risk factors. Notably, long-term heavy coffee drinking created a vicious cycle between neck and shoulder pain and sleep duration of less than 6 h. The implication of the findings is that individuals who sleep less than 6 h, or who suffer from neck and shoulder pain, should limit their coffee intake to two cups per day.

## Figures and Tables

**Figure 1 jpm-13-00025-f001:**
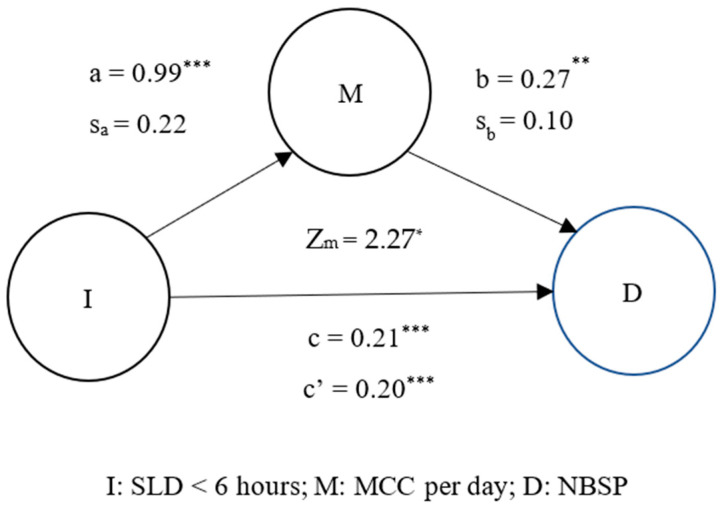
Mediation effect of MCC per day on SLD < 6 h and NBSP. ** p* < 0.05; *** p* < 0.01; **** p* < 0.0001; M, mediation factor; I, independent variable; D, dependent variable.

**Figure 2 jpm-13-00025-f002:**
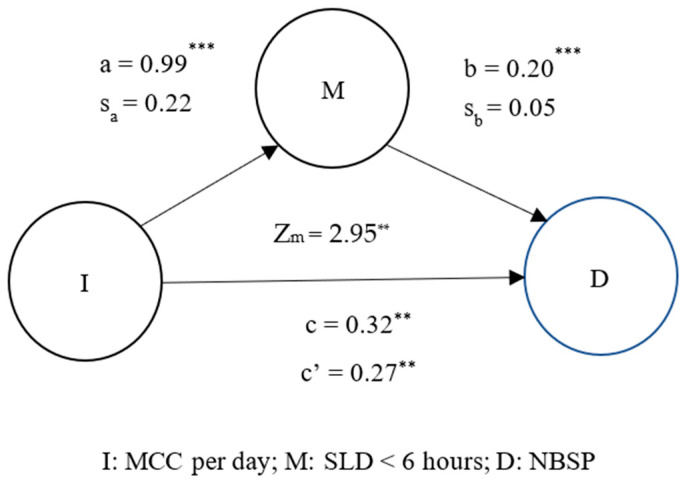
Mediation effect of SLD < 6 h on MCC per day and NBSP. *** p* < 0.01; **** p* < 0.0001; M, mediation factor; I, independent variable; D, dependent variable; c is the simple linear or logistic regression coefficient for the independent variable against dependent variable in the absence of mediation factor; c’ is the binary linear or logistic regression coefficient for the independent variable against dependent variable in the presence of mediation factor; a is the simple linear or logistic regression coefficient for the independent variable against the mediation factor; b is the regression coefficient for the mediation factor against the dependent variable in the binary linear or logistic regression model; $s_{a}$ and $s_{b}$ represent the standard deviations of a and b.

**Figure 3 jpm-13-00025-f003:**
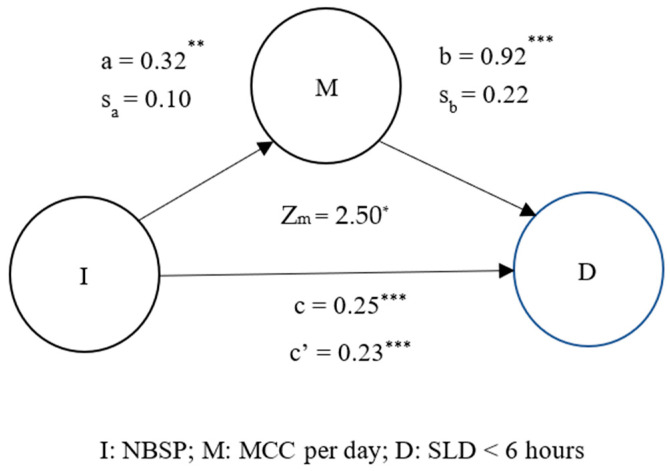
Mediation effect of MCC per day on NBSP and SLD < 6 h. ** p* < 0.05; *** p* < 0.01; **** p* < 0.0001; M, mediation factor; I, independent variable; D, dependent variable; c is the simple linear or logistic regression coefficient for the independent variable against dependent variable in the absence of mediation factor; c’ is the binary linear or logistic regression coefficient for the independent variable against dependent variable in the presence of mediation factor; a is the simple linear or logistic regression coefficient for the independent variable against the mediation factor; b is the regression coefficient for the mediation factor against the dependent variable in the binary linear or logistic regression model; $s_{a}$ and $s_{b}$ represent the standard deviations of a and b.

**Table 1 jpm-13-00025-t001:** MS pain sites and factor analysis of the NMQ.

MS Pain Sites	N	%	Score	Factor Loading	
			Mean ± SD	Factor 1	Factor 2
Neck	585	36.22	26.76 ± 37.64	0.33	−0.02
Left shoulder	325	20.12	15.07 ± 31.62	0.33	−0.01
Right shoulder	371	22.97	17.64 ± 33.89	0.33	0.02
Upper back	273	16.90	12.90 ± 29.77	0.17	0.00
Waist or lower back	451	27.93	20.20 ± 34.72	0.08	−0.04
Left elbow	70	4.33	3.29 ± 16.26	−0.05	−0.04
Right elbow	113	7.00	5.33 ± 20.43	−0.04	−0.04
Left wrist	77	4.77	3.72 ± 17.38	−0.05	0.00
Right wrist	162	10.03	7.51 ± 23.66	−0.03	−0.03
Left hip/thigh/buttock	67	4.15	3.12 ± 15.64	−0.05	−0.07
Right hip/thigh/buttock	68	4.21	3.17 ± 15.83	−0.02	−0.04
Left knee	80	4.95	3.78 ± 16.98	−0.05	−0.07
Right knee	88	5.45	4.17 ± 18.05	−0.02	−0.04
Left ankle	29	1.80	1.26 ± 10.10	−0.02	0.49
Right ankle	25	1.55	1.10 ± 9.58	−0.02	0.54
		Eigenvalues		4.93	1.55
		Explained variation %		57.59	18.12

N, individuals; %, the proportion of individuals suffering from MS pain.

**Table 2 jpm-13-00025-t002:** Differences in the frequency of pain among the survey variables.

		Score on the Frequency of Musculoskeletal Pain			
			Mean ± SD		
Survey Variables	Individuals	NBSP Score	*p*-Value	BAP Score	*p*-Value
**Gender**					
Female	1314	0.04 ± 0.93	<0.001 ^a^	−0.01 ± 0.85	0.643 ^a^
Male	301	−0.17 ± 0.84		0.02 ± 0.90	
**Age**					
Less than or equal to 29	412	−0.11 ± 0.86	0.003 ^b^	−0.00 ± 0.81	0.420 ^b^
Between 29 and 38	433	0.01 ± 0.90		−0.06 ± 0.41	
Between 38 and 45	302	0.15 ± 0.96		0.04 ± 1.01	
More than or equal to 45	468	−0.01 ± 0.95		0.03 ± 1.06	
**Marriage**					
Married	779	0.07 ± 0.96	0.003 ^a^	−0.02 ± 0.79	0.330 ^a^
Other	836	−0.07 ± 0.87		0.02 ± 0.91	
**Having children**					
Parents	703	0.07 ± 0.96	0.006 ^a^	−0.00 ± 0.88	0.914 ^a^
Not parents	912	−0.06 ± 0.88		0.00 ± 0.84	
**Engaging in leisure activities with family/friends**					
Always	102	−0.05 ± 0.89	0.601 ^b^	0.00 ± 0.78	0.764 ^b^
Often	498	−0.04 ± 0.92		0.03 ± 0.96	
Sometime	765	0.03 ± 0.94		−0.02 ± 0.77	
Seldom	238	0.02 ± 0.87		0.02 ± 0.93	
Never	12	−0.21 ± 0.59		−0.18 ± 0.33	
**Education**					
Master’s degree or above	297	0.11 ± 0.99	0.034 ^a^	−0.10 ± 0.24	<0.0001 ^a^
University degree or below	1318	−0.03 ± 0.90		0.02 ± 0.94	
**Sleep duration per day**					
Less than 5 h	63	0.26 ± 1.04	<0.001 ^b^	0.29 ± 1.88	0.069 ^b^
Between 5 and 6 h	563	0.12 ± 1.00		0.01 ± 0.91	
Between 6 and 7 h	719	−0.06 ± 0.85		−0.04 ± 0.68	
Between 7 and 8 h	232	−0.14 ± 0.87		0.01 ± 0.82	
More than 8 h	38	−0.10 ± 0.75		0.01 ± 0.46	
**Coffee intake per day**					
More than 2 cups per day	26	0.61 ± 1.25	<0.001 ^b^	−0.17 ± 0.20	0.853 ^b^
2 cups per day	70	0.18 ± 0.97		−0.03 ± 0.62	
1 cup per day	556	0.06 ± 0.95		−0.01 ± 0.88	
Occasionally	678	−0.04 ± 0.90		0.02 ± 0.82	
Never	285	−0.13 ± 0.81		−0.00 ± 0.97	
**Alcohol use**					
Alcohol use in a month	609	0.10 ± 0.97	0.001 ^a^	0.01 ± 0.84	0.857 ^a^
No alcohol use in a month	1006	−0.06 ± 0.88		−0.00 ± 0.86	
**Exercise at least once a week**					
Yes	933	−0.07 ± 0.87	0.001^a^	−0.01 ± 0.80	0.705 ^a^
No	682	0.09 ± 0.97		0.01 ± 0.93	
**Overtime work in a month**					
More than 80 h	5	0.54 ± 1.35	<0.001 ^b^	−0.25 ± 0.41	0.587 ^b^
45–80 h per month	54	0.44 ± 1.14		0.04 ± 1.46	
Fewer than 45 h	502	0.09 ± 0.96		0.04 ± 1.46	
Seldom	1054	−0.07 ± 0.87		−0.02 ± 0.73	
**Shift schedules**					
Irregular shifts	192	0.16 ± 1.05	0.075 ^a^	−0.06 ± 0.49	0.445 ^a^
Regular shifts	196	−0.04 ± 0.91		0.02 ± 0.88	
Night shifts	166	−0.05 ± 0.84		0.08 ± 1.13	
Day shifts	1061	−0.15 ± 0.91		−0.01 ± 0.85	
**Profession**					
Physicians	138	0.03 ± 1.01	0.036 ^b^	−0.01 ± 0.80	0.889 ^b^
Nurses	613	0.08 ± 0.94		0.02 ± 1.00	
Professional and technical personnel	283	−0.06 ± 0.84		−0.02 ± 0.59	
Administrative staff	581	−0.06 ± 0.90		−0.01 ± 0.82	
**Suffering from chronic diseases**					
Yes	638	0.20 ± 1.03	<0.0001 ^a^	0.04 ± 1.16	0.195 ^a^
No	977	−0.13 ± 0.81		−0.03 ± 0.57	

Note: SD, standard deviation; ^a^
*t* test; ^b^ one-way ANOVA; NBSP, neck and both shoulders pain; BAP, both ankles pain.

**Table 3 jpm-13-00025-t003:** Effect of SLD and drinking MCC per day on NBSP.

	Unstandardized Linear Regression Coefficient (B) for NBSP					
	Simple Regression			Multiple Regression^1^		
Main Effect	B	SE	*p*	B	SE	*p*
Drinking MCC per day	0.32	0.10	0.001	0.23	0.10	0.016
SLD < 6 h per day	0.21	0.05	<0.0001	0.15	0.05	0.001

SE, standard error; B, unstandardized linear regression coefficient; ^1^ model was in the presence of adjusted variables, including gender, age, marriage, having children, education, alcohol use, exercise, overtime work, profession, and suffering from chronic diseases.

## Data Availability

The original contributions presented in this study are included in the article/Appendix A. Further inquiries can be directed to the corresponding author.

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
