# Peer review of "The Effects of Frequent Coffee Drinking on Female-Dominated Healthcare Workers Experiencing Musculoskeletal Pain and a Lack of Sleep"

_jpm, 2022, doi:10.3390/jpm13010025_

Round 1

Reviewer 1 Report

The present study examines the effect of frequent coffee consumption (2 cups per day) on individuals suffering from musculoskeletal pain and lack of sleep. It examines a total of 3 hypotheses: a) Coffee intake is significantly associated with an increased risk of musculoskeletal pain b) People with shorter sleep duration are more prone to musculoskeletal pain c) Coffee intake may lead to the vicious cycle between musculoskeletal pain and lack of sleep.

The manuscript is well written and structured but there are points that need to be clarified. The authors' research hypotheses are scientifically valid, and the experimental design is appropriate to test them. Also, the results seem to be in agreement and supported by the existing literature. Ethics statements as well as data availability statements appear to be sufficient.

Τhe references of the last five years are 16, while the rest are previous ones. There are references that need to be corrected. Pages (Ref.n.  14, 20, 35 and 38) and missing doi (Ref. n. 10,27,31, 32,33, 35,41 and 43) should be added.

 The statistical analysis requires some clarifications:

At the beginning, in the results paragraph, a series of p-values is being reported, without mentioning the statistical test used to derive them.

It is unclear when it is used univariate or multivariate analysis.

Ιn table 3 is indicated as univariable instead of univariate.

It would be necessary to mention in figures that a, b, c, are coefficients for the corresponding model. Although there is some general information at the end of the Materials and methods paragraph, in Figures 1.1, 1.2, 1.3 shows c and c΄, without accompanying their corresponding explanation.

Especially in fig.1.3 it is unclear when it is used univariate or multivariate analysis.

Also, in Figure 1.3, the title does not match the model and the order of the variables is incorrect. 1st is MCC, 2ndis NBSP, and the 3rd is SLD<6.

Pag 14…(Table 3, B=0.22, P=0.016). 0.22 must be replaced by 0.23

Pag 14…«more caffeine per day than those who reported a sleep duration of less than 6 hours». «Less» must be replaced by«more»

On page 11, it is mentioned twice, should instead of shoulder pain

Please explain why it is mentioned in the heading "period of COVID-19" and in the rest of the manuscript there is no mention.

Author Response

Suggestions for revision 1-1

The references for the last five years are 16, while the rest are previous ones. There are references that need to be corrected. Pages (Ref. n.  14, 20, 35, and 38) and missing doi (Ref. n. 10, 27, 31, 32, 33, 35, 41, and 43) should be added.

Response 1-1

Thanks to the reviewer for the comments. We have revised all Ref. except Ref. n. 10 and 41 because of there is no doi.

Suggestions for revision 1-2

The statistical analysis requires some clarifications:

At the beginning, in the results paragraph, a series of p-values is being reported, without mentioning the statistical test used to derive them.

Response 1-2

Thanks to the reviewers for comments. We have added two pieces of Supplementary information Table S2 and S3 (The description of sleep duration and coffee intake per day for all individuals) to fix the unclear problems for the statistical test.

Suggestions for revision 1-3

It is unclear when it is used univariate or multivariate analysis.

Ιn table 3 is indicated as univariable instead of univariate.

Response 1-3

Thanks to the reviewers for comments. We changed two titles in Table 3 as simple regression and multiple regression to clearly indicate the original meaning. All "univariate" had been replaced by "simple" in the text.

Suggestions for revision 1-4:

It would be necessary to mention in the figures that a, b, and c, are coefficients for the corresponding model. However, there is some general information at the end of the Materials and methods paragraph, in Figures 1.1, 1.2, and 1.3 show c and c΄, without accompanying their corresponding explanation.

Response 1-4

Thanks to the reviewers for comments. We have added the new supplementary information for a,b,c, and c’ in Figures 1.1, 1.2, and 1.3.

The new paragraphs followed:

c, the simple linear or logistic regression coefficient for the independent variable against the dependent variable in the absence of mediation factor; c’, the binary linear or logistic regression coefficient for the independent variable against the dependent variable in the presence of mediation factor;  is the simple linear or logistic regression coefficient for the independent variable against the mediation factor;  is the regression coefficient for the mediation factor against the dependent variable in the binary linear or logistic regression model; represent the standard deviation of and.

Suggestions for revision 1-5:

Especially in fig.1.3 it is unclear when it is used univariate or multivariate analysis.

Response 1-5

Thanks to the reviewers for comments. Figure 1.1~Figure 1.3 adopted the formulas developed by Iacobucci (2012) [52] to determine if the mediation effect existed. (I-M) It is the simple linear or logistic regression for the independent variable (I) against the mediation factor (M), while (M-D) It is binary linear or logistic in presence of an adjusted variable (mediation factor).

Suggestions for revision 1-6:

Also, in Figure 1.3, the title does not match the model and the order of the variables is incorrect. 1st is MCC, 2ndis NBSP, and the 3rd is SLD<6.

Response 1-6:

Thanks to the reviewers for comments. The title had been renewed as “Figure 1.3. Mediation effect of MCC per day on NBSP and SLD < 6 hours”. (Line 250)

Suggestions for revision 1-7:

Page 14…(Table 3, B=0.22, P=0.016). 0.22 must be replaced by 0.23

Response 1-7

Thanks to the reviewers for comments. We have fixed the mistake in Line 291 (Table 3, B=0.23, P=0.016).

Suggestions for revision 1-8:

Pag 14…«more caffeine per day than those who reported a sleep duration of less than 6 hours». «Less» must be replaced by«more»

Response 1-8:

Thanks to the reviewers for comments. We had fixed the mistake in Line 317 as “…those who reported a sleep duration of more than 6 hours.”

Suggestions for revision 1-9:

On page 11, it is mentioned twice, should instead of shoulder pain

Response 1-9:

Thanks to the reviewers for comments. We had fixed two mistakes in Line 214 and Line 222 as “shoulder pain”.

Suggestions for revision 1-10:

Please explain why it is mentioned in the heading "period of COVID-19" and in the rest of the manuscript, there is no mention.

Response 1-10:

Thanks to the reviewers for comments. Because we collect our data during the COVID-19 pandemic and we don’t make sure if the results are impacted due to the pandemic, we add the words "COVID-19 pandemic" in my title. That explanation will be added to the research limitation.

Line 342-345: Notably, we were unable to determine whether high work stress or emotional exhaustion due to the pandemic affected the findings, so a similar study during the non-pandemic period should be replicated and compared with the result of the pandemic period.

Reviewer 2 Report

The paper addresses an interesting issue, but it is hardly limited by several methodological factors. Moreover, the discussion needs a complete revision.

These are the most important to me:

1. In introduction You speak about MS pain, but myocardial infarction can  not be considered muscoskleletan pain, and maybe it is related to coffee induced tachycardia.

2. In introduction, sleep problems can reduce pain tolerance, and pain can induce insomnia but there is not cause-effect relationship as the sentence suggests.

3. You say that insomnia is common in medical staff, but this concept as no relation to your study.  

4. Once again, you speak about acupuncture but therefore no mention was made.These 2 sentences make no sense for me.

5.No mention of cover in introduction. It is your population and you didn't speak of it in all introduction.

6. In your material and methods section you should just explain inclusion and exclusion criteria and how your study was conducted, the number of patients included should be reported in results.Moreover, in results section you should reported age, sex, and describe your included population.

7. In discussion, you report that several evidences support hypothesis 1, but the evidence is 1 and it referred to fibromyalgia, that , as you know, is not a MS pain condition.

8. In result section, no mention was made about educational level, pain, work and so on. All these factors are related in my opinion, to MS pain, sleep deprivation and also coffee intake and they are ignored in your results and discussion. Even if your hypotheses are  clearly stated and you focus on them, you should not completely ignore other factors that could also interfere with pain , sleep and also coffee intake.

9. It is a little bit surprising that exercise increase pain, and your reference (21) seems to negate this result. How do you explain this contradictory results?

10. In title the COVID period was mentioned, but how this factor influences your research, if an influence could be searched, is not further analysed. So why do you mention it? How does COVID influence your result?

10. Table 2 is not clear at all.

11. Why did you shift coffee intake and therefore put different categories together? It was just a statistical method to increase your number?Was it justified with  any objective reason?

11. All methods section does not explain how these patients were enrolled, when , who enrolled them and why.

Author Response

Suggestions for revision 2-1:

  1. In introduction You speak about MS pain, but myocardial infarction can not be considered muscoskleletan pain, and maybe it is related to coffee induced tachycardia.

Response 2-1

Thanks to the reviewers for comments. "Myocardial infarction" mainly corresponds “health” part. To clearly indicate, we rewrite our original paragraph as followed:

L8-14: Additionally, despite clinical studies demonstrating caffeine’s adjuvant analgesic effects [5], long-term coffee drinking can have a negative effect on health and musculoskeletal (MS) pain. In health, individuals drinking more than 5 cups of coffee per day can have an increased risk of myocardial infarction or unstable angina [6]. In MS pain, related research has shown that drinking more than 7 cups of coffee per day was associated with a higher risk of knee osteoarthritis among Korean males [7].

Suggestions for revision 2-2:

  1. In introduction, sleep problems can reduce pain tolerance, and pain can induce insomnia but there is not cause-effect relationship as the sentence suggests.

Response 2-2:

Thanks to the reviewers for comments. We had changed the causal relationship to a close relationship (Line 35) after checking the original paragraph. That more expresses the original meaning of the article.

Suggestions for revision 2-3:

  1. You say that insomnia is common in medical staff, but this concept as no relation to your study.  

Response 2-3:

Thanks to the reviewers for comments. We had adjusted my paragraphs and references' order in Lines 28-40 to better explain the original meaning.

Poor sleep quality is a common health problem among medical staff [28,29]. Reduced sleep duration and poor sleep quality in individuals have become more common during the past decades [30], leading to poor health outcomes [31] and even increased mortality [32]. Despite the recommended minimum sleep duration of 7 hours per night for healthy adults, only 25% of adults achieve this amount [33]. Notably, lack of sleep can lead to day functional impairment [34], increased occupational injury [35], and reduced productivity [36].

Overall, there is a close relationship between sleep and MS pain. For instance, since sleep problems can significantly reduce pain tolerance [37], individuals suffering from chronic pain are more likely to experience insomnia [38]. Caffeine in coffee can also reduce total sleep duration, efficiency, and quality [39]. In addition, frequent consumption of caffeinated drinks can have a negative impact on habitual sleep duration [40], too.

Suggestions for revision 2-4:

  1. Once again, you speak about acupuncture but therefore no mention was made. These 2 sentences make no sense for me.

Response 2-4:

Thanks to the reviewers for comments. We had adjusted some paragraphs and sentences in Lines 47-51 to further explain the original meaning of the article.

Line 47-51: “In this regard, one study of mice found that acupuncture causes nucleotides and adenosine to release and to relieve pain [46]. Yet, these antinociceptive effects but can be blocked by caffeine [47]. Notably, individuals who suffer from chronic insomnia disorder have been associated with reduced adenosine [48]. Moreover, impaired sleep significantly increases the risk of reduced pain tolerance [39], too.”

Suggestions for revision 2-5:

  1. No mention of cover in introduction. It is your population and you didn't speak of it in all introduction.

Response 2-5

Thanks to the reviewers for comments. We had added Supplementary information Table S1, S2, and S3 to further describe demographic.

Suggestions for revision 2-6:

  1. In your material and methods section you should just explain inclusion and exclusion criteria and how your study was conducted, the number of patients included should be reported in results. Moreover, in results section you should reported age, sex, and describe your included population.

Response 2-6:

Thanks to the reviewers for comments. We had adjusted some paragraphs and rewrite sentences to strengthen the introduction of materials and methods. Please confirm the new paragraph in lines 64-69.

In addition, we added three Supplementary information for demography. Table S2. The description of sleep duration per day for all individuals; Table S3. The description of coffee intake per day for all individuals.

Suggestions for revision 2-7:

  1. In discussion, you report that several evidences support hypothesis 1, but the evidence is 1 and it referred to fibromyalgia, that , as you know, is not a MS pain condition.

Response 2-7:

Thanks to the reviewers for comments. Fibromyalgia is a common disorder whose cardinal manifestation is chronic, widespread pain (Clauw DJ. Fibromyalgia: a clinical review. JAMA. 2014;311:1547-55). Musculoskeletal pain is the most prominent feature of Fibromyalgia, it is useful to focus on the features of the pain that can help distinguish it from other disorders (Daniel J. Clauw, Fibromyalgia: An Overview, 2009, The American Journal of Medicine,122 (12), Issue 12: S3-S13).

Suggestions for revision 2-8:

  1. In result section, no mention was made about educational level, pain, work and so on. All these factors are related in my opinion, to MS pain, sleep deprivation and also coffee intake and they are ignored in your results and discussion. Even if your hypotheses are clearly stated and you focus on them, you should not completely ignore other factors that could also interfere with pain, sleep, and also coffee intake.

Response 2-8

Thanks to the reviewers for comments. As shown in Table 2, The factors that affect on MS pain such as gender, age, marriage, having children, education, alcohol use, exercise, overtime work, profession, and suffering from chronic diseases, etc. In addition, we also cited the past studies' results to support our findings (Line 269-272). Therefore, the multiple linear regression was adopted for adjusting or controlling these effects on MS pain. Overall, we determined drinking MCC per day and SLD <6 hours per day were associated with increased NBSP in the presence of adjusting risk factors. As shown in Table 3, both are independent risk factors of NBSP, respectively.

Suggestions for revision 2-9:

  1. It is a little bit surprising that exercise increase pain, and your reference (21) seems to negate this result. How do you explain this contradictory results?

Response 2-9:

Thanks to the reviewers for comments. We had fixed the mistake for exercise habit in Line 175 as no weekly exercise (mean±SD= 0.09 ± 0.97). The result is consistent with reference 21.

Suggestions for revision 2-10:

  1. In title the COVID period was mentioned, but how this factor influences your research, if an influence could be searched, is not further analyzed. So why do you mention it? How does COVID influence your result?

Response 2-10

Thanks to the reviewers for comments. Because we collect our data during the COVID-19 pandemic and we don’t make sure if the results are impacted due to the pandemic, we add the words "COVID-19 pandemic" in my title. Those explanations will be added to the research limitation.

Line 342-345: Notably, we were unable to determine whether high work stress or emotional exhaustion due to the pandemic affected the findings, so a similar study during the non-pandemic period should be replicated and compared with the result of the pandemic period.

Suggestions for revision 2-11:

  1. Table 2 is not clear at all.

Response 2-11

Thanks to the reviewers for comments. We have improved the explanation in Table 2. Please confirm it.

Suggestions for revision 2-12:

  1. Why did you shift coffee intake and therefore put different categories together? It was just a statistical method to increase your number? Was it justified with any objective reason?

Response 2-12

Thanks to the reviewers for comments. Because of too little sample size (N=26) for drinking more than 2 cups of coffee per day, we must through merging two categories of drinking 1 cup of coffee and more than 2 cups of coffee per day for increasing the sample size to overcome the insignificant problem on the statistic.

Suggestions for revision 2-13:

  1. All methods section does not explain how these patients were enrolled, when , who enrolled them and why.

Response 2-13

Thanks to the reviewers for comments. We had added some paragraphs and sentences in materials and methods to further explain the process and methods of recruiting participants.

Line 64-69: This observational and cross-sectional study initially was conducted at a hospital affiliated with a medical university in Taichung, Taiwan, from March to April 2021. All 2531 healthcare workers who have served for one year in the hospital were distributed a QR code of google forms linked questionnaires by email. 1,633 (64.52%) individuals completed the self-reported questionnaire, after which 1,615 (63.81%) were deemed valid after those with missing data were excluded.

Round 2

Reviewer 2 Report

The paper is significantly improved, and I would like to congrats with the authors.

However, I still do not understand your research., above all from title.

COVID-19 does not change your results (or you can  not investigate this point)so why it is in title? I mean does it matter? Does it add scientific sound?

Moreover, your sample is made by female (81%) hospital workers, but no mention about it nor in title, nor in discussion and in limitation.

This is a critical point, that limits your results and applicability and you simply ignore these data.Why female experienced more pain? It is work load, parental load, it is sexual predisposition, different female pain perception and reporting?What about other workers? Also the division in nurse and physician   can shows some results (in other tables you divide degree or university degree and this separation is not clear to me:  does a technical personal for example belong to what category?)

Maybe a paper  called "The Effect of Frequent Coffee Drinking on female hospital workers  Experiencing Musculoskeletal Pain and Lack of Sleep" will be more appropriate.

Interestingly, only 4% of patients enrolled reported  they drink coffee >2 times/day, but almost all patients reported pain... how do you explain these results? Is statistically and clinically justified to draw conclusion from only 4% of your population?

The same papers you cite use other cut off (more than 4) that is very different from your results: how do you justify your results? It is a quite different cut off that is completely ignored in your discussion. Does it make any differences?How do you explain the different effect in your population?

In the discussion section I still do not understand If exercise improves or worsens pain...it seems that it increases pain, but in results you present completely different data.

English needs extensive editing since many sentences are not clear at all (for example:in discussion the first paragraph).

The statistical methods is very strong and the interplay between coffee, sleep deprivation and pain is intriguing but I still think that this paper requires a profound revision and does not address the question in the right way.

Author Response

The paper is significantly improved, and I would like to congrats with the authors.

However, I still do not understand your research, above all from title.

Suggestions for revision 1

COVID-19 does not change your results (or you can not investigate this point) so why it is in title? I mean does it matter? Does it add scientific sound?

Response 1:

Thanks to the reviewer for the comments. We will change our title to “The Effect of Frequent Coffee Drinking on Female-Dominated Healthcare Workers Experiencing Musculoskeletal Pain and Lack of Sleep”

Suggestions for revision 2

Moreover, your sample is made by female (81%) hospital workers, but no mention about it nor in title, nor in discussion and in limitation.

Response 2

Thanks to the reviewer for the comments. We will change our title to “The Effect of Frequent Coffee Drinking on Female-Dominated Healthcare Workers Experiencing Musculoskeletal Pain and Lack of Sleep”

Suggestions for revision 3

This is a critical point, that limits your results and applicability and you simply ignore these data. Why female experienced more pain? It is work load, parental load, it is sexual predisposition, different female pain perception and reporting? What about other workers? Also the division in nurse and physician   can shows some results (in other tables you divide degree or university degree and this separation is not clear to me:  does a technical personal for example belong to what category?)

Response 3

Thanks to the reviewer for the comments. We have added paragraphs in the Discussion further introducing this important limitation.

Line 344-351

Despite the adjustment for sex in the multiple linear regression, the results of the present study could be better suitable to women because female participants accounted for >80% of the study population. Regarding the sex difference in MS pain, it could be caused by estrogen and progesterone. For instance, testosterone, the major male sex hormone, protects men from chronic MS pain [60]. Because the study population only included physicians, nurses, professional and technical personnel, and administrative staff, we added “healthcare workers” in the title to limit the applicability to occupational groups.

Suggestions for revision 4

Maybe a paper called "The Effect of Frequent Coffee Drinking on female hospital workers Experiencing Musculoskeletal Pain and Lack of Sleep" will be more appropriate.

Response 4

Thanks to the reviewer for the comments. We will change our title to “The Effect of Frequent Coffee Drinking on Female-Dominated Healthcare Workers Experiencing Musculoskeletal Pain and Lack of Sleep”

Suggestions for revision 5

Interestingly, only 4% of patients enrolled reported they drink coffee >2 times/day, but almost all patients reported pain... how do you explain these results? Is statistically and clinically justified to draw conclusion from only 4% of your population?

Response 5

Thanks to the reviewer for the comments.   Table S2 in Supplementary information indicated individuals who drink 2 cups (N= 70) or drink more than 2 cups of coffee (N= 26) were over 60% (61.46%) of those who experience sleep duration was less than 6h. These results could hint poor sleep, long-term drinking of multiple cups of coffee, and neck and shoulder pain could exist in close relationships. We adopted mediation analysis to test the relationship. According to mediation models (Figure 1.1 and Figure 1.2), we determined drinking more than 2 cups of coffee per day and shorter sleep form a vicious circle that further worsens neck and shoulder pains.

Suggestions for revision 6

The same papers you cite use other cut off (more than 4) that is very different from your results: how do you justify your results? It is a quite different cut off that is completely ignored in your discussion. Does it make any differences? How do you explain the different effect in your population?

Response 6

Thanks to the reviewer for the comments. The caffeine’s effect on MS pain reaches the threshold depending on one’s genetics. Therefore, no matter whether 4-12 cups/day or over 7 cups/day is a relative size. The threshold of Asians on caffeine could be relatively low. However, the over 2 cups of coffee intake per day are relatively high among our participants. We have added a paragraph in limitation (Line 334-336) to further introduce.

Line 334-336: In addition, caffeine’s effect on MS pain reaches the threshold depending on one’s genetics [56]. Therefore, the threshold of >2 cups of coffee per day might not be suitable for other countries or race.

Suggestions for revision 7

In the discussion section I still do not understand If exercise improves or worsens pain...it seems that it increases pain, but in results you present completely different data.

Response 7

Thanks to the reviewer for the comments. A cross-sectional study of MS pain among 10,000 adults from the general working population (cited 21) found a high level of physical activity (i.e. 5 h per week) was associated with a lower risk of low back pain and neck-shoulder pain. Another study of MS pain in Norway (N= 77,216) demonstrated a relatively small amount of physical exercise (i.e., 1–1.9 hours/week) lowers the risk of chronic pain in the low back and neck/shoulders (om Ivar Lund Nilsen, Andreas Holtermann, Paul J. Mork, Physical Exercise, Body Mass Index, and Risk of Chronic Pain in the Low Back and Neck/Shoulders: Longitudinal Data From the Nord-Trøndelag Health Study, American Journal of Epidemiology, 174(3), 2011, 267–273). Our study found individuals who report exercise at least once a week significantly sustained low levels of neck and shoulder pain (p= 0.001). This is an interesting issue that is worth to further exploring in the future study.

English needs extensive editing since many sentences are not clear at all (for example: in discussion the first paragraph).

Response 8

Thanks to the reviewer for the comments. We have edited by an English native.

The statistical methods is very strong and the interplay between coffee, sleep deprivation and pain is intriguing but I still think that this paper requires a profound revision and does not address the question in the right way.

Response 9

Thanks to the reviewer for the comments. We have revised profoundly throughout.